# Experimental Investigation on Compressive Strength, Ultrasonic Characteristic and Cracks Distribution of Granite Rock Irradiated by a Moving Laser Beam

**Lianfei Kuang [1]**, **Lipeng Sun [1]**, **Dongxu Yu [1]**, **Yijiang Wang [1],*[ID]**, **Zhaoxiang Chu [1]** and **Jo Darkwa [2]**

[1] State Key Laboratory for Geomechanics and Deep Underground Engineering, School of Mechanics and Civil Engineering, China University of Mining and Technology, Xuzhou 221116, China

[2] Faculty of Engineering, University of Nottingham, University Park, Nottingham NG7 2RD, UK

* Correspondence: yjwang@cumt.edu.cn

**Abstract:** Efficient fracturing is the key issue for the exploitation of geothermal energy in a Hot Dry Rock reservoir. By using the laser irradiation cracking method, this study investigates the changes in uniaxial compressive strength, ultrasonic characteristics and crack distributions of granite specimens by applying a laser beam under various irradiation conditions, including different powers, diameters and moving speeds of the laser beam. The results indicate that the uniaxial compressive strength is considerably dependent on the power, diameter and moving speed of the laser beam. The ultrasonic-wave velocity and amplitude of the first wave both increase with a decreased laser power, increased diameter or moving speed of the laser beam. The wave form of irradiated graphite is flattened by laser irradiation comparing with that of the original specimen without laser irradiation. The crack angle and the ratio of the cracked area at both ends are also related to the irradiation parameters. The interior cracks are observed to be well-developed around the bottom of the grooving kerf generated by the laser beam. The results indicate that laser irradiation is a new economical and practical method that can efficiently fracture graphite.

**Keywords:** thermal irradiation; laser cracking; mechanical property; ultrasonic characteristic; crack distribution

## 1. Introduction

The geothermal energy in a Hot Dry Rock (HDR) reservoir is usually stored in a deep graphite stratum with a depth of about 3–10 km [1].It has a significant advantage over other energy sources [2]. The exploitation and utilization of HDR geothermal energy have attracted great interest. It has been estimated that the United States' total HDR geothermal resources is equivalent to about $1.4 \times 10^{25}$ J [3]. The prospective source bases for enhanced geothermal systems in Great Britain and Germany were assessed and the potential economic power generation by geothermal energy was deduced to be 223 and 447 $GW_e$, respectively [4,5]. Recently, the utilization of geothermal energy showed a rising trend in China, with a geothermal-power-generation capacity of 27.78 $GW_e$ in 2014 [6,7].

Due to the lower porosity and permeability properties, the exploitation of HDR geothermal energy comes at a high cost and technical difficulty [8]. Many researchers are investigating efficient granite-cracking measures to build a deep-HDR geothermal energy reservoir. A considerable amount of research has been conducted to investigate the propagation of natural and artificial fractures in HDRs using hydraulic fracking [9–11]. It was found that geothermal energy or oil/gas production can be increased by hydraulic fracturing [12–14]. Hydraulic fracturing has been widely regarded as one of the most effective means. However, induced seismicity was often observed due to the application of a high-pressure fluid injection, which can affect the application of HDR geothermal power plants [15–17]. In addition, hydraulic fracturing also has some deficiencies, including

high-initiation pressure, pore plugging in the water-sensitive stratum and groundwater-contamination risk [18]. In order to avoid these problems caused by hydraulic fracturing, a series of waterless fracturing measures were developed by researchers to crack the hard rock, such as gas fracturing [19,20], liquid-nitrogen fracturing [21–23], thermal spallation fracturing [24,25], microwave fracturing [26–29] and laser fracturing [30–34].

Among these new fracturing measures, the high-power laser-beam solution has excellent potential. It can be used to crack hard rock to improve the fracturing efficiency and reduce construction costs. Li et al. [31,32] studied the stress variation and fragment mechanism of granite generated by laser perforation both experimentally and numerically. They found that the ultimate tensile stress achieved by laser irradiation ranged from 481 to 536 MPa, which was much greater than that of the granite specimen. Wang et al. [33,34] experimentally investigated the temperature change in and mechanical properties of rock specimens generated by laser irradiation. They found that the temperature gradient of irradiated material can reach 5000 °C/mm, which led to a significant reduction in the compressive strength. Hu et al. [30] studied the concrete perforation by using a continuous $CO_2$ laser. They found that the maximum rate of perforation for wet specimens was greater than that for dry concrete. Jurewicz [35] employed a $CO_2$ laser machine to study the effect of laser power and moving speed on the penetration depth and volumetric rock-removal rate. It was found the laser kerf showed a good fracturing efficiency for hard-rock excavations. In a study by Ahmadi et al. [36], the specific energy and perforating hole depth were investigated. They reported that the specific energy of water-saturated rock was the highest compared with the oil-saturated and dry specimens. A 6 kW fiber laser was employed by Kariminezhad et al. [37] to investigate the laser's perforation characteristic. It was found that the perforation efficiency was not dependent on the rock size and deposition orientation when the boundary effect was not considered. Erfan et al. [38] employed a long-pulse Nd:YAG laser to examine the impact of the moving perforation of rock and they reported that the perforating efficiency of moving perforation was greater than that of non-moving perforations. Ndeda et al. [39] also studied the thermal stress of granite caused by pulsed-laser spallation. The results demonstrate that residual stress is high, and sudden cooling after laser irradiation can also contribute to increased crack propagation. Buckstegge et al. [40] found that the formation of crack and splintering caused by laser irradiation was due to irregular thermal expansion. Yang et al. [41,42] experimentally investigated the rock-temperature distribution, specific energy and rate of perforation. They found that the rock-temperature profile strongly depended on the laser power and rock composition. Bharatish et al. [43] employed a $CO_2$ laser with a power output of 12 kW to drill rock specimens to study the impact of various laser parameters on the drilling characteristics. The results show that the rate of perforation and specific energy are related to the laser-power output and irradiation time. Similar conclusions were reported by Yan et al. [44] who investigated laser perforation on rocks.

It can be concluded that a considerable amount of research was conducted on the highly efficient exploitation of geothermal energy in HDR reservoirs by using hydraulic fracturing and some new waterless fracturing methods. However, the previous investigations mainly focused on laser fracturing, drilling efficiency and perforation rate. A study on the mechanical property, ultrasonic characteristic and crack distribution in laser-irradiated granite using a moving beam has not yet been reported. Therefore, on the basis of granite irradiation experiments, the influences of laser-power output, beam diameter and laser-beam moving speed on the mechanical property, ultrasonic characteristic and crack distribution of irradiated granite specimens are experimentally studied in this paper to provide some insights into those aspects.

## 2. Methodology

### 2.1. Granite Specimen

The cylindrical specimens with diameter lengths of φ50 mm × 100 mm were used based on the China Standard GB/T50266-2013. Both ends of the granite specimens were

ground flat to increase the accuracy of mechanical-property measurements. The uniaxial compressive strength of granite ranged between 120 and 140 MPa, with the main minerals of quartz and albite. The detailed technical data were presented in our previous paper [33].

### 2.2. Experimental Devices

Figure 1 shows the main experimental setup in this study.

(1) Laser system: The maximum power of the continuous fiber laser was 1 kW. The laser was transferred from the laser device to the cutting head, which was fixed in an industrial robot with six axles, as shown in Figure 1a. The robot was responsible for the movement of the laser cutting head.

(2) Electro-hydraulic testing servo machine: An electro-hydraulic testing servo machine (loading capacity: 2000.0 kN) with an accuracy of ±1% (at the full scale) was used, as shown in Figure 1b.

(3) Ultrasonic detector: The ultrasonic tests of rock specimens were conducted by using an ultrasonic detector with a voltage amplitude accuracy of ±3.5% and a time measurement accuracy of ±0.5%, as shown in Figure 1c.

(4) X-ray micro-imaging system: The crack distribution of the irradiated specimen was reconstructed by using the three-dimensional X-ray micro-imaging system (3D-XRM), as presented in Figure 1d.

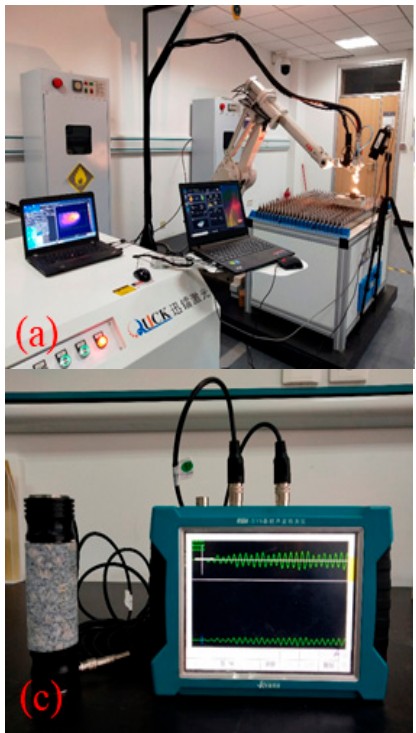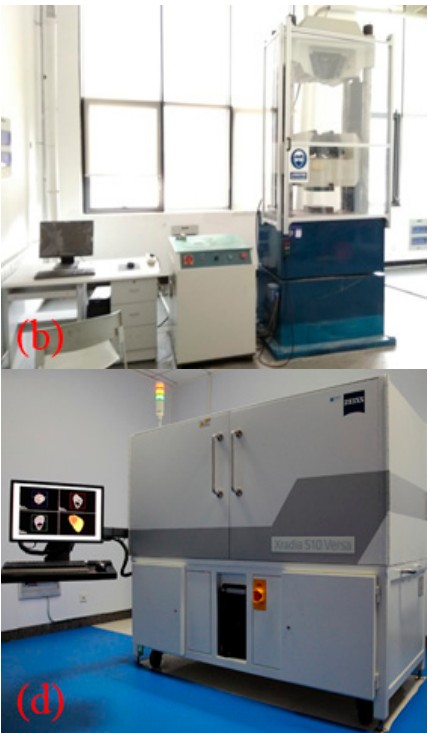

**Figure 1.** Devices used in experiments: (**a**) laser system, (**b**) electro-hydraulic testing servo machine, (**c**) ultrasonic tester, (**d**) X-ray micro-imaging system.

In addition, the main components and their contents of the specimen were tested through XRD (X-ray diffraction) and XRF (X-ray fluorescence).

### 2.3. Experimental Program and Data Processing

This study aimed to investigate the influence of laser power, beam diameter and moving speed of the laser beam on the uniaxial compressive strength, ultrasonic characteristic and crack distribution of irradiated granite specimens. Because the total thermal energy irradiated to the rock specimen can be significantly influenced by the laser-power output and irradiation time, a series of experiments based on the laser power ranging from

400–1000 W, the beam diameter ranging from 6–12 mm and the moving speed ranging from 0.5–4 mm/s were conducted, as listed in Table 1.

**Table 1.** Experimental scheme.

| Parameters | I | II | III | IV |
| --- | --- | --- | --- | --- |
| Laser power (W) | 400 | 600 | 800 | 1000 |
| Laser-beam diameter (mm) | 6 | 8 | 10 | 12 |
| Moving speed of laser beam (mm/s) | 0.5 | 1.0 | 2.0 | 4.0 |

The laser beam with various irradiation parameters moved from left to right in parallel to the specimen under a fixed speed controlled by a robot, as shown in Figure 2.

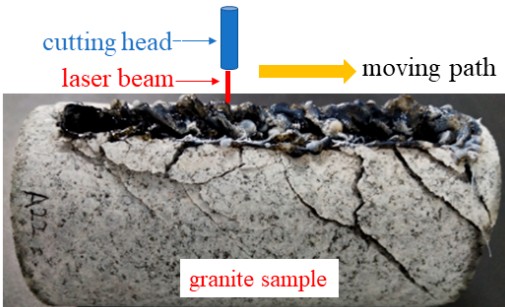

**Figure 2.** Schematic of moving laser irradiation.

The testing accuracy and uncertainty were analyzed to ensure the accuracy of the experimental results. The equations of uncertainties for directed variables, including rock mass and length, are present by [45]:

$$u_{\mathrm{v}} = \sqrt{\Delta_{\mathrm{v}}^2 + \sigma_{\mathrm{v}}^2} \tag{1}$$

where $u_{\mathrm{v}}$ is the uncertainty of the directed variables, $\Delta\mathrm{v}$ is the testing accuracy and $\sigma_{\mathrm{v}}$ is the Bessel equation of the standard deviation, which is shown in the following Equation:

$$\sigma_{\mathrm{v}} = \sqrt{\dfrac{\sum\limits_{i}^{N}(x_i - \overline{x})^2}{N - 1}} \tag{2}$$

where $x_i$ and $\overline{x}$ are individual testing values and the mean value of individual testing values, respectively, $N$ is the number of testing items.

The equations of uncertainties for undirected variables (including the volume and uniaxial compressive strength) are shown as [45]:

$$u_{\mathrm{v}}' = \sqrt{\sum\limits_{i}^{n}\left(\dfrac{\partial F}{\partial x_i} \cdot \Delta_{x_i}\right)^2} \tag{3}$$

$$u_{\mathrm{v}}' = \sqrt{\sum\limits_{i}^{n}\left(\dfrac{\partial \ln(F)}{\partial x_i} \cdot \Delta_{x_i}\right)^2} \tag{4}$$

where $u_{\mathrm{v}}' = F(x_i)$ is the undirected variable calculated from $x_i$. Equation (3) could be used to compute uncertainties if $F(x_i)$ just includes the operators of addition and subtraction. Equation (4) should be employed if $F(x_i)$ just includes the operators of multiplication and division.

The uncertainties of variables are calculated, as shown in Table 2. The measurement uncertainty of uniaxial compressive strength was about 2–5%, which indicates that the

testing accuracy can be ensured. The standard deviation of the uniaxial compressive strength, ultrasonic-wave velocity and amplitude of the first wave were also presented to quantify their divergence.

**Table 2.** Testing accuracy and uncertainty.

| Variables | Mass | Length | Volume | Compressive Strength |
|---|---|---|---|---|
| Testing accuracy | $\pm$0.01 g | $\pm$0.02 mm | - | - |
| Uncertainties | 2~5% | 0.11% | 0.08% | 2–5% |

## 3. Results

### 3.1. Mechanical Properties

#### 3.1.1. Uniaxial Compressive Strength

The influences of laser-power output, beam diameter and beam moving speed on the uniaxial compressive strength of irradiated rock specimens were investigated, and the results are presented in Figure 3. As shown in Figure 3a, the increased laser-power output results in a significant reduction in the compressive strength. For instance, the compressive strength nonlinearly decreases from 135 MPa of the original specimen without laser irradiation (original specimen) to 93, 83, 69 and 54 MPa, respectively, when the laser-power output increases from 0 W to 400, 600, 800 and 1000 W (diameter and moving speed of laser beam were fixed at 6 mm and 0.5 mm/s, respectively). The compressive strength of the irradiated specimens was reduced by about 31.1%, 38.7%, 48.8% and 60.0%, respectively, compared to that of the original specimen. This may be attributed to the fact that the higher-power laser beam can generate more thermal energy and also result in greater thermal damage to the rock specimens. The uniaxial compressive strength was therefore reduced with an increased laser power. However, the beam diameter had a negative effect on the thermal damage of the irradiated granite specimens, as shown in Figure 3b. The uniaxial compressive strength was increased from 54 to 84 MPa when the laser-beam diameter increased in the range of 6–12 mm (laser power and moving speed of laser beam were fixed at 1000 W and 0.5 mm/s, respectively). The uniaxial compressive strength of the specimen irradiated by a 12 mm laser beam was only reduced by 38% compared to that of the original specimen. The effect of the beam's moving speed on the uniaxial compressive strength is shown in Figure 3c. The uniaxial compressive strength was decreased from 130 to 54 MPa when the laser beam's moving speed was reduced from 4–0.5 mm/s (laser power and beam diameter were fixed at 1000 W and 6 mm, respectively), in which the uniaxial compressive strength was slowly reduced to 101 MPa when the moving speed decreased to 1 mm/s. However, the uniaxial compressive strength of the irradiated specimen was significantly reduced to 54 MPa when the moving speed was reduced further to 0.5 mm/s. Therefore, this suggests that a lower moving speed should be used to achieve greater thermal damage in the granite. As can be seen from the profiles presented in Figure 3, all these influences of laser power, laser-beam diameter and laser-beam moving speed on the uniaxial compressive strength of irradiated specimens are non-linear. The stress–strain curve for the original specimen and irradiated granite with different irradiation parameters are presented in Figure 3d. One can observe that the compression distance of the irradiated specimens is longer than that of the original rock. The strains corresponding to the peak stress of the irradiated specimens are dependent on the irradiation parameters, which are larger than that of the original one.

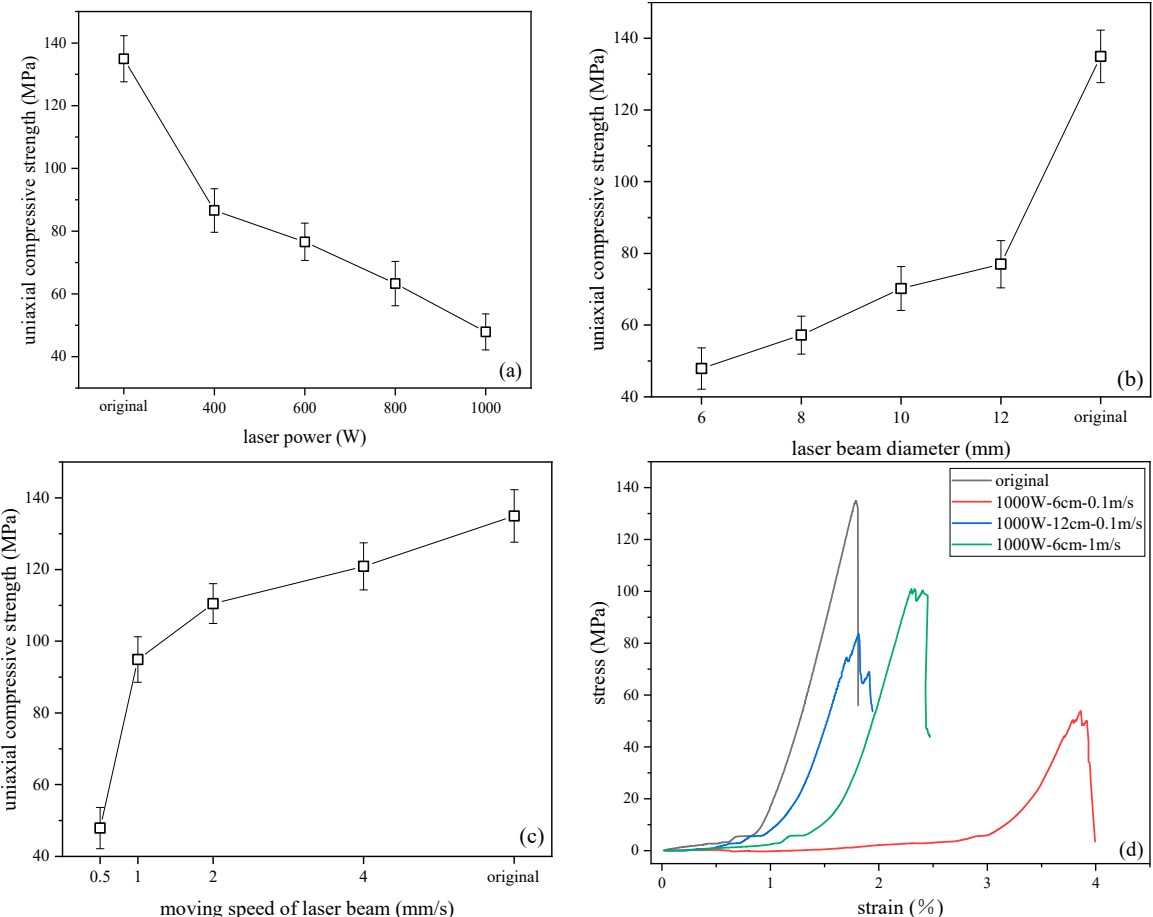

**Figure 3.** Uniaxial compressive strength versus (**a**) laser power, (**b**) laser-beam diameter, (**c**) moving speed of laser beam, (**d**) uniaxial compressive strength–strain curve.

When the uniaxial compression stress approached the peak, the granite instantaneously fractured accompanied by sounds similar to the rock burst. A considerable number of cracks penetrating from the top to bottom were observed when the uniaxial compression load approached the yield strength of the original rock specimen, as shown in Figures 3d and 4. A typical splitting failure with a slight deformation can be confirmed for the original granite exposed to uniaxial compression. This can contribute to the fact that the granite is a kind of rock with high brittleness. The same cracks can be seen from the irradiated specimens shown in Figure 4. However, the deformations considerably increased when the laser-power increased, and a detailed discussion is presented in Section 3.1.2. In addition, the desquamation and fragments falling from the irradiated specimens were observed during the uniaxial-compressive-strength test, which was due to the thermal damage caused by the laser irradiation. Figure 4 also indicates that the failure modes of irradiated specimens are similar under various irradiation conditions. The destruction of the rock specimen was attributed to the internal microcracks' initiation, propagation and penetration under the uniaxial compressive load.

The rock temperature rapidly increased due to the laser irradiation. For instance, the temperature-increasing rate of limestone irradiated by the laser beam can approach 14 °C/ms [34], which demonstrates that laser irradiation could be classified as a thermal shock process. Owing to the lower thermal conductivity of granite, the thermal energy inside the rock specimen cannot be rapidly transferred, which resulted in the high temperature gradient near the perforated hole with values of about 2669–5700 °C /mm [33]. Therefore, the high thermal stress induced by the high temperature gradient could lead to considerable thermal damage to the granite specimens. The numerical results show that

the thermal stress generated by the laser irradiation range from 481–536 MPa [32], which is far beyond their own intensity of the granite specimen. In addition, the high temperature could also contribute to the breakage of many metallic bonds, including Al-O, K-O, Na-O and Fe-O. Certain amounts of microdefects in the granite specimen were observed [46].

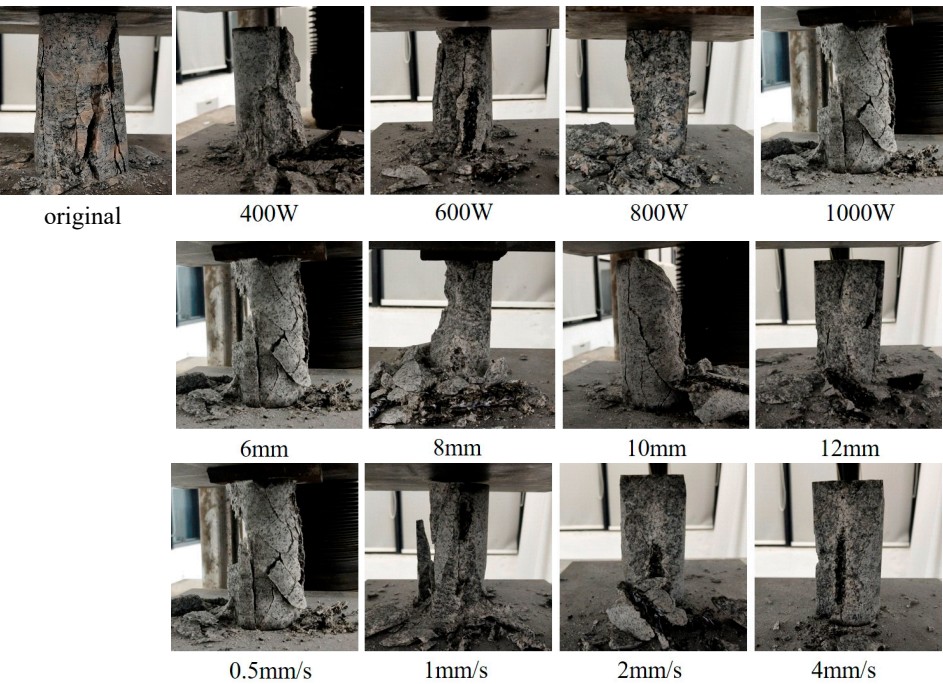

**Figure 4.** Images of granite with different irradiation parameters under uniaxial compression.

### 3.1.2. Component Analysis

XRD analysis was used to determine the mineralogical changes in the original and irradiated rock specimens, as shown in Figure 5. According to the XRD results presented in Figure 5a, the main mineral of the original granite consists of calcium-sodium feldspar, potassium feldspar, quartz and biotite, with the molten temperatures of 1215 °C, 1290 °C, 1713 °C and 1800 °C, respectively. As shown in Figure 5b, quartz and pyroxene are the main minerals of the molten graphite. The diffraction intensity of molten graphite is considerably reduced compared with that of the original graphite specimen, which indicates that the crystalline material is transformed into an amorphous state due to the high temperature generated by laser irradiation.

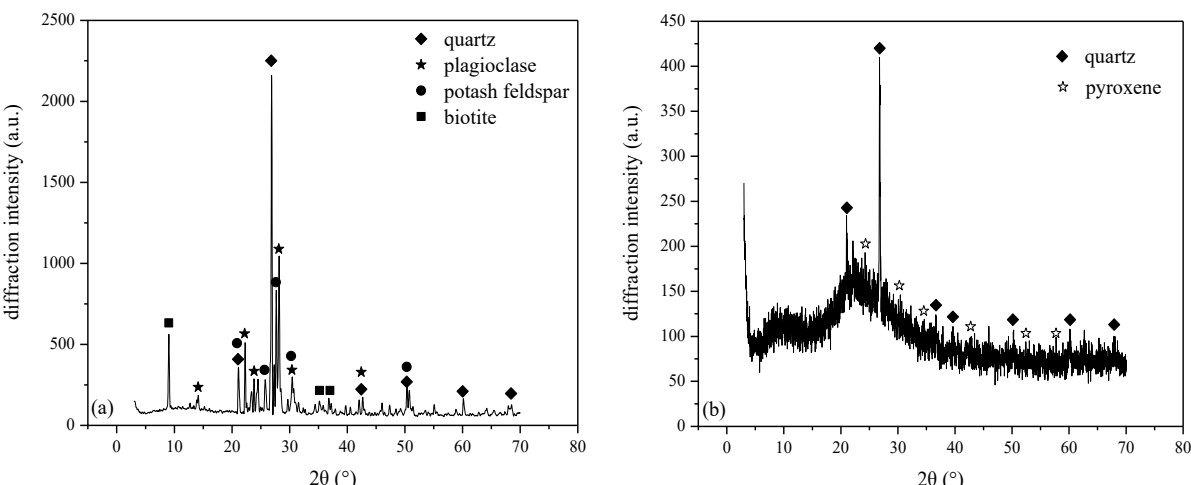

**Figure 5.** XRD curve of (**a**) original specimen and (**b**) molten graphite.

The XRF results listed in Table 3 demonstrate that the main component of the molten graphite is similar to that of the original rock. It is also found from the table that the content of $SiO_2$ of the irradiated specimen slightly increases by 1.22% (from 68.55% to 69.77%); meanwhile, the content of other compositions is slightly decreased comparing with that of the original specimen.

**Table 3.** Composition of original and irradiated granite (%).

| Mineral | $Na_2O$ | MgO | $Al_2O_3$ | $SiO_2$ | $K_2O$ | CaO | $Fe_2O_3$ |
|---|---|---|---|---|---|---|---|
| Original | 3.53 | 0.69 | 13.91 | 68.55 | 5.16 | 1.63 | 2.46 |
| Irradiated | 3.19 | 0.50 | 12.82 | 69.77 | 4.31 | 1.58 | 2.36 |

*3.2. Ultrasonic Characteristics*

3.2.1. Wave Velocity and Amplitude

The effect of various irradiation conditions on the ultrasonic-wave velocity and first-wave amplitude is investigated and presented in Figure 6. The ultrasonic-wave velocity was about 4.45 km/s for the original specimen, with the penetrating time of 22.5 μs. However, the wave velocity decreased from 3.89 to 2.92 km/s for the irradiated specimens when the laser power ranged between 400 to 1000 W, with the penetrating time ranging from 24.9 to 32.5 μs, as shown in Figure 6a. Compared with the value of the original specimen, the wave velocity of the irradiated rock decreased by 14.5–35.8% when the laser power increased in the range of 400–1000 W. In addition, the first-wave amplitude was also observed to be decreased under the same range of the laser-power output. For instance, the first-wave amplitude decreased from 134.25 (original specimen) to 102.1 dB (1000 W), with a reduction rate of 23.9%. The influence of the beam diameter on the wave velocity and the first-wave amplitude is illustrated in Figure 6b. It is seen from the figure that the wave velocity increases from 3.18 to 3.75 km/s when the beam diameter increases in the range of 6–12 mm, with an increasing rate of 18.0%. Similarly, an increase in the beam diameter with the same range also leads to an increased wave amplitude from 102.1 to 129.65 dB. The effect of moving speed on the wave velocity and the first-wave amplitude is presented in Figure 6c, which indicates that the wave velocity increases from 3.18 to 4.24 km/s when the moving speed is increased in the range of 0.5–4.0 mm/s. It is also found that the first-wave amplitude increases from 102.1 to 125.95 dB under the same range of the moving speed. In addition, the increasing rate of the wave velocity and first-wave amplitude is also observed to be significantly weakened when the laser beam moves faster. Figure 6 also indicates that all these influences of laser-power output, beam diameter and beam moving speed on the wave velocity and first-wave amplitude of the irradiated specimens are non-linear.

It is believed that the ultrasonic-wave velocity is strongly related to the elastic characteristic and uniaxial compressive strength of the rock specimen [47]. When the ultrasonic wave encounters microcracks filled with air on the propagation path, it is time-consuming for the ultrasonic wave to penetrate the rock–air interface. The ultrasonic wave propagates along the edges of the cracks, which indicates that the time for the ultrasonic wave to penetrate the irradiated rock with microcracks is longer than that for the original specimen. Therefore, the cracks within the irradiated rock specimen caused by laser irradiation can lead to a longer penetrating time and a lower wave velocity. When the ultrasonic wave penetrates those rock specimens consisting of microcracks or sub-regions with a lower strength, an increase in the absorption and scattering attenuation could be observed, which results in a lower wave amplitude [48]. The cracks and defects within the irradiated specimens are equivalent to the obstacles that can interfere with the normal transmission of the ultrasonic wave. The variation of the first-wave amplitude is strongly dependent on the compactness of the rock specimens, which is more sensible than the penetrating time. Therefore, the more the cracks and defects, the larger the attenuation of the first-wave amplitude is [49]. Because the interference from other waves' superpositions may affect the validity of the results, the first-wave amplitude was used for the comparison in this study.

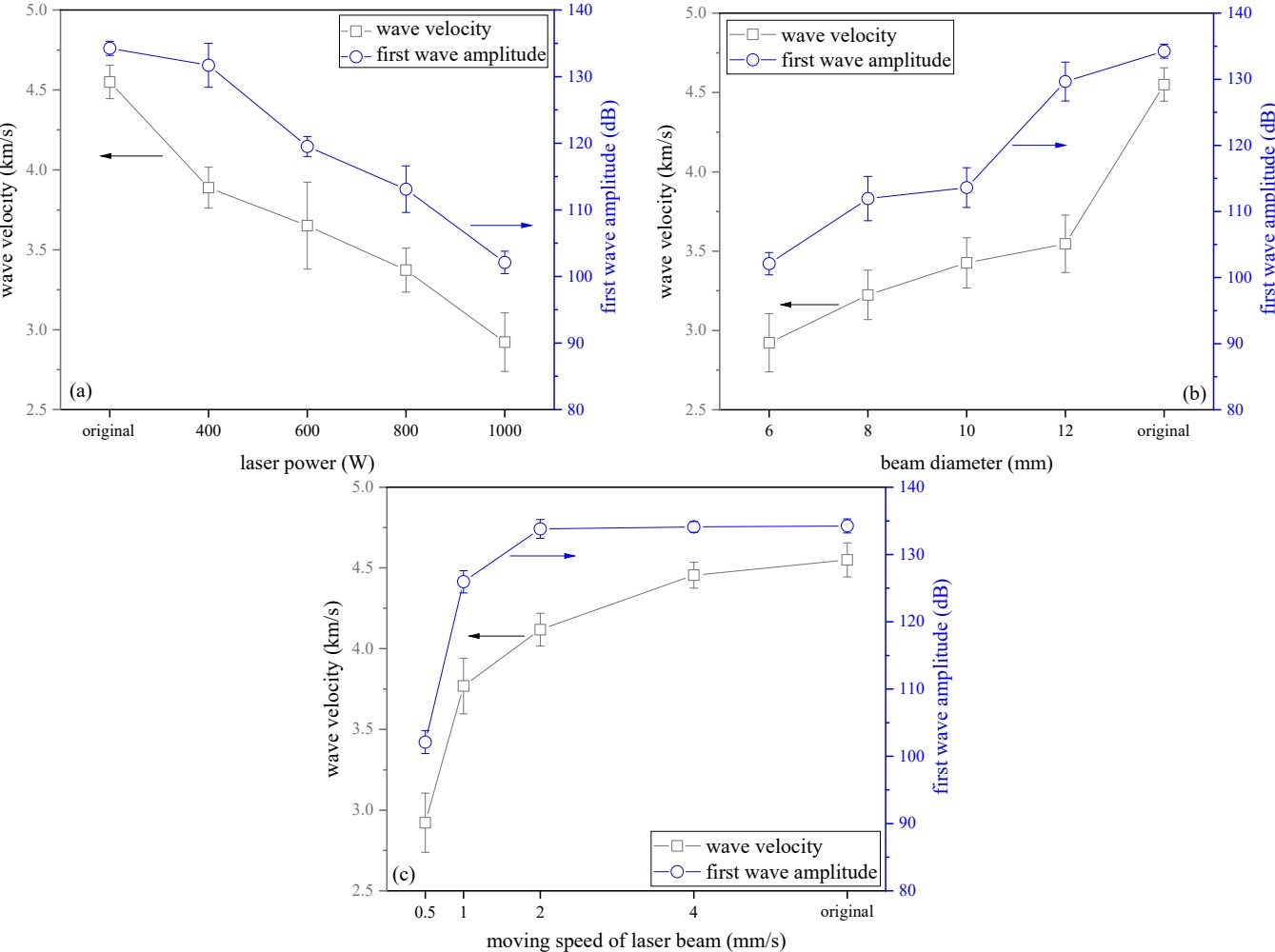

**Figure 6.** The influence of (**a**) laser-power output, (**b**) beam diameter, (**c**) moving speed on wave velocity and first-wave amplitude.

### 3.2.2. Wave Form

The influence of various irradiation parameters on the wave form was also investigated and presented in Figure 7. The original rock-specimen case is shown in Figure 7a, from which a perfect sinusoid shape can be observed. It is indicated that the original rock specimen is homogenous without any cracks or defects inside the graphite specimen. Figure 7b,c illustrate the wave form of the irradiated specimen under the irradiation power outputs of 600 and 1000 W, respectively. It is seen that the wave form of the irradiated specimen is distorted and the voltage amplitude of the first wave considerably reduces from 15.63 to 2.38 mV when the laser power increases in the range of 600–1000 W. The effect of the beam diameter on the wave form is shown in Figure 7d,e, which demonstrates that the voltage amplitude of the first wave increases from 11.38 to 105.63 mV when the beam diameter increases in the range of 8–12 mm. The influence of the laser-beam moving speed on the wave form is shown in Figure 7f,g. It is seen that the voltage amplitude of the first wave increases from 47.50 to 110.25 mV when the moving speed increases in the range of 1–4 mm/s.

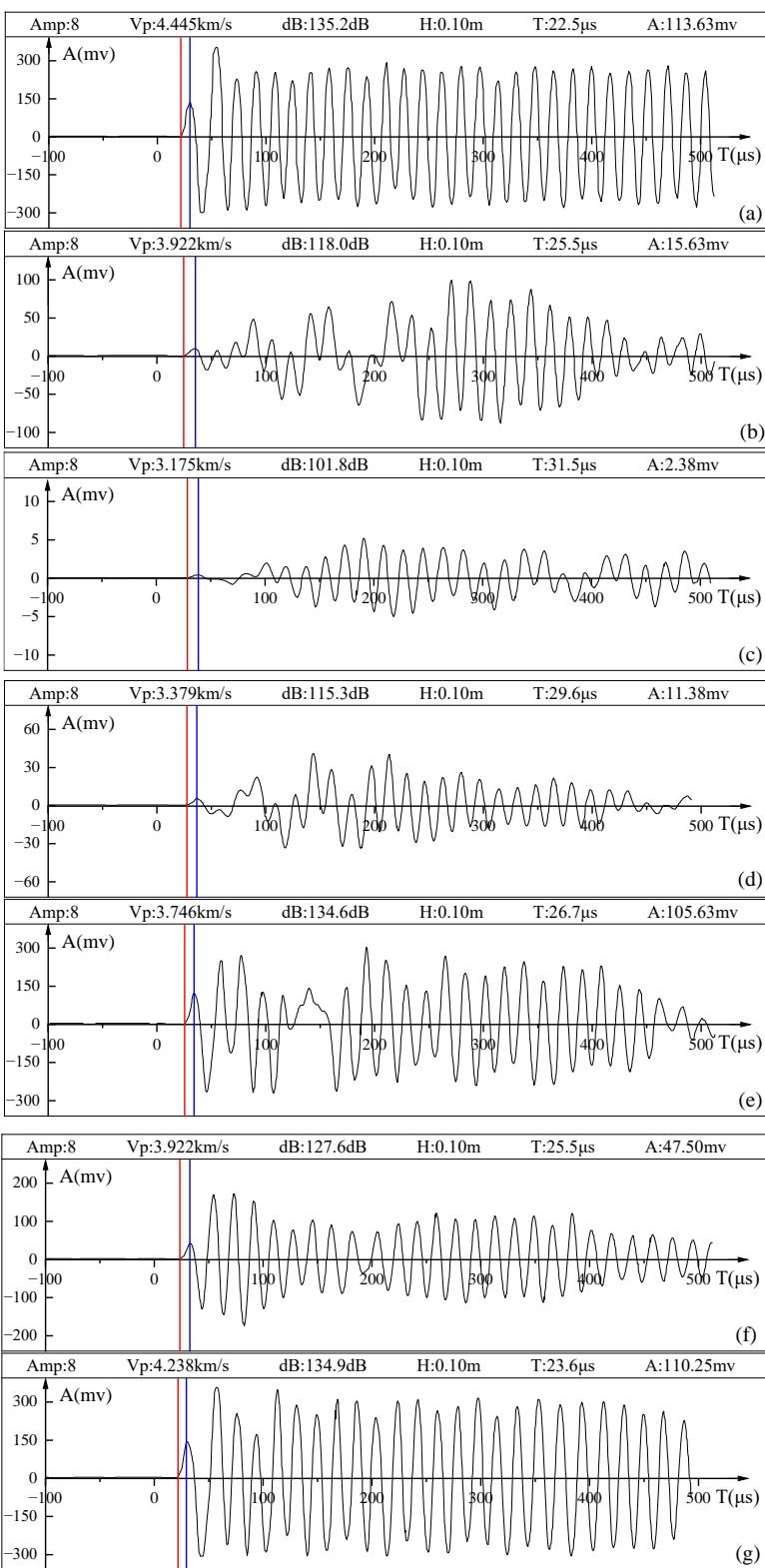

**Figure 7.** Wave form versus (**a**) original specimen, laser powers of (**b**) 600 W and (**c**) 1000 W, beam diameters of (**d**) 8 mm and (**e**) 12 mm and moving speeds of (**f**) 1 mm/s and (**g**) 4 mm/s.

The wave form is significantly flattened by the increased laser-power output, decreased beam diameter and beam moving speed. Compared with the waveform of the original specimen presented in Figure 7a, the wave forms of irradiated specimens are distorted because the granite specimens are thermally damaged with cracks and defects generated by

the laser beam. The reason for the variation in the wave form is attributed to the reflection and refraction of the ultrasonic wave caused by the fractures' interfaces, which leads to the formation of shear waves and phase superposition of the ultrasonic wave.

### 3.3. Cracks Distribution

3.3.1. Surface Cracks

The distribution of surface cracks generated by the laser beam was studied after the rock specimens were naturally cooled. The images of the irradiated specimens are presented in Figures 8 and 9, from which obvious cracks can be observed on the surface of each irradiated rock specimen. In addition, the detailed number and angle of cracks caused by laser irradiation were also investigated, as shown in Figure 10.

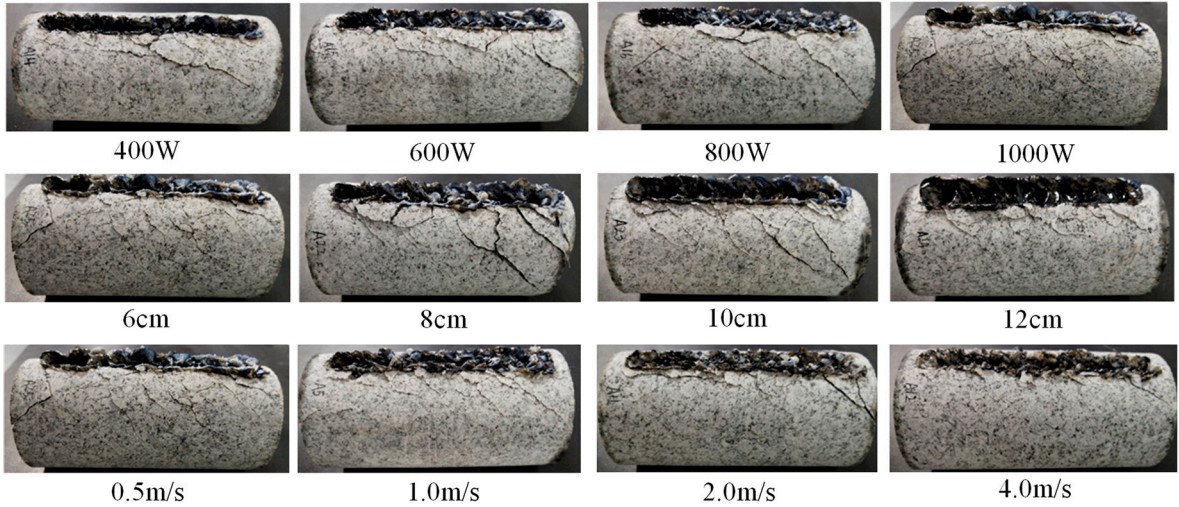

**Figure 8.** Cracks of irradiated granite on the lateral surface.

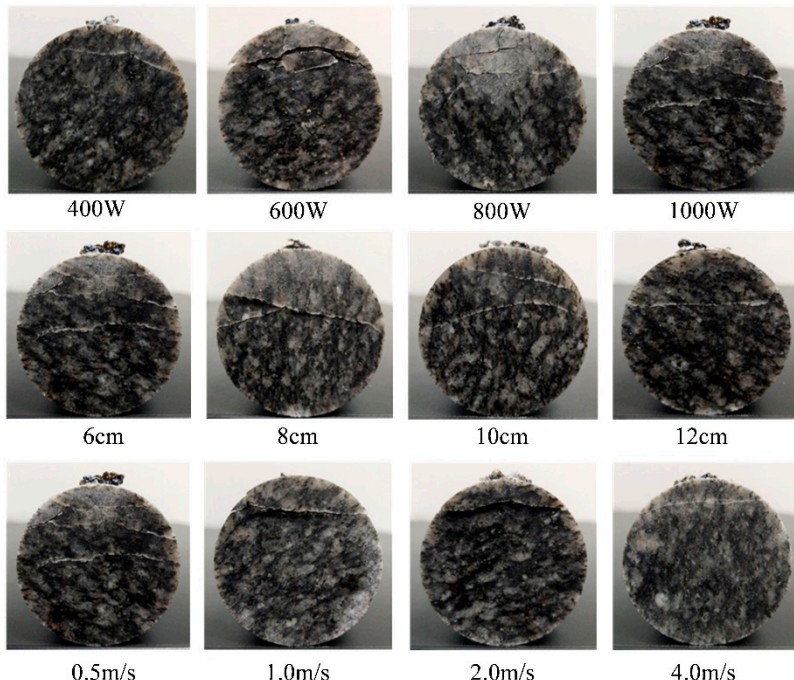

**Figure 9.** Cracks of irradiated granite on the bottom surface.

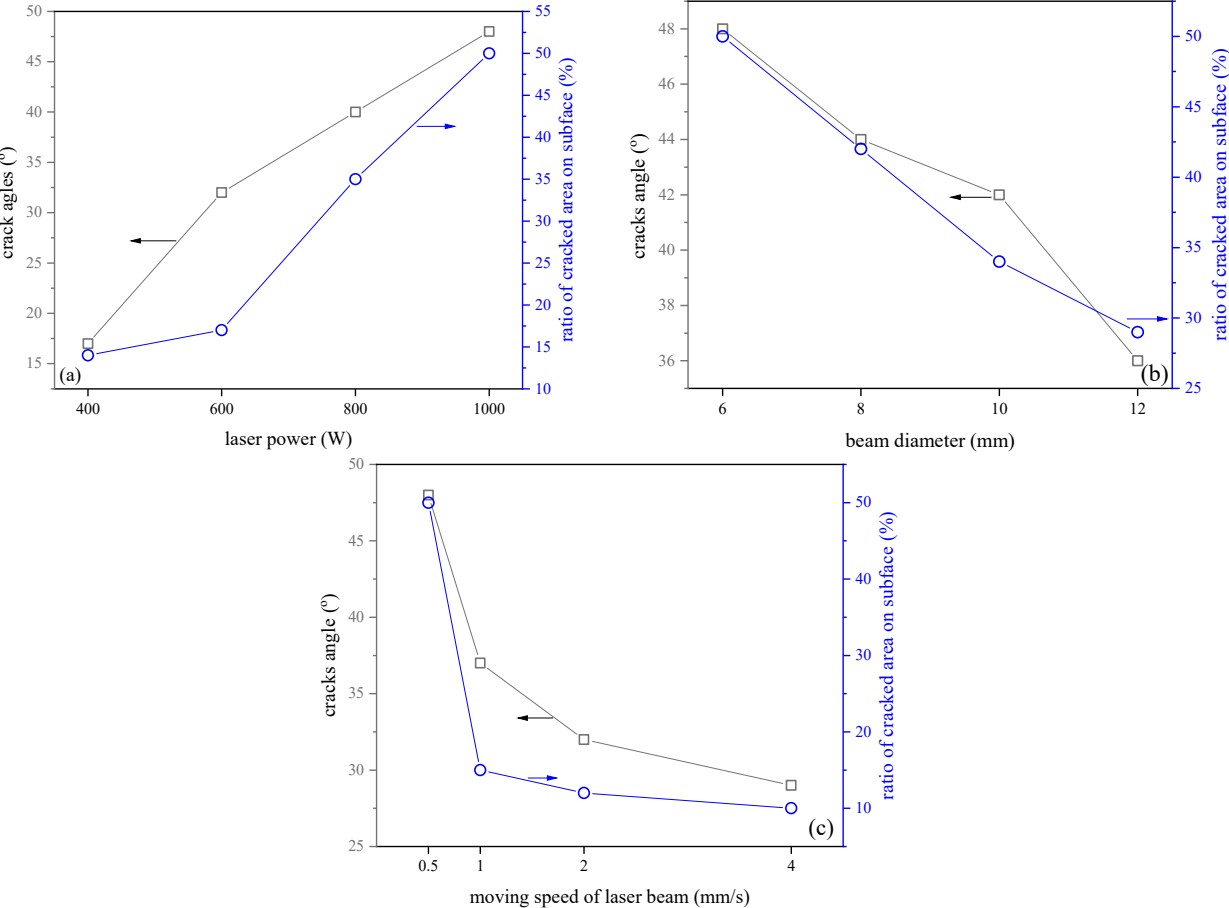

**Figure 10.** The influence of (**a**) laser power, (**b**) beam diameter, (**c**) moving speed of laser beam on crack angles and ratio of cracked area.

　　One can observe that the crack numbers and angles both increase from 3 to 7 and 17° to 48°, respectively, when the laser power increases in the range of 400–1000 W, as shown in Figure 10a. The crack numbers and angles increase by 133% and 182%, respectively. The ratio of the cracked area at both ends also increases from 14% to 50% when the laser power increases in the range of 400–1000 W, which illustrates that a higher-power laser beam results in a larger cracked body. This may be attributed to the high-temperature field with a super-large temperature gradient achieved by the laser irradiation when the laser power increases, thereby causing more thermal damage [34]. However, Figure 10b shows that an increase in the laser-beam diameter can lead to a decrease in the crack angle. For instance, the crack angle decreased from 48° to 36° when the beam diameter increased in the range of 6–12 mm. Interestingly, it was also found that the ratio of the cracked area at both ends decreased from 50% to 29% under the same range of beam diameter, which was attributed to the decreased power density on the irradiation surface. It led to less thermal damage when a larger beam diameter was used [33]. Finally, Figure 10c shows that the crack angle reduces when the beam's moving speed increases, given the same laser power and beam diameter. For instance, an increase in the beam's moving speed from 0.5–4 mm/s led to a decrease in the crack angle from 48° to 29°. In addition, the ratio of cracked area at both ends was also reduced from 50% to about 10% due to the increased moving speed in the range of 0.5–4.0 mm/s. This is understandable considering that less time was required to achieve the same irradiation length when the beam moving speed was increased. Therefore, less thermal damage to the granite specimen can be expected because less thermal energy was irradiated to the specimen.

### 3.3.2. Interior Cracks

3D X-ray micro-imaging (3D-XRM) was applied to a selected irradiated rock specimen to obtain the detailed interior structure and crack distributions. CT scanning technology can visually present the microstructure's characteristics (such as cracks, pores and microcracks) of the irradiated rock by using 256 gray scales through the density difference of each imaging unit in the specimen. Then, Dragonfly was applied for post-processing, and data, such as crack sizes inside the irradiated sample, were obtained by means of smooth filtering, threshold segmentation, mesh reconstruction and size measurement. Only images from the case with the laser-power output of 1000 W, beam diameter of 6 mm and beam moving speed of 0.5 m/s were presented because similar damage to irradiated specimens and the interior crack distributions could be expected in the other cases.

Figure 11 shows the rendered images of the irradiated granite specimen and its interior cracks, which qualitatively illustrate the crack distributions within and on the surface of the granite specimen after laser irradiation. From the front view of the irradiated specimen shown in Figure 11a, one can see that the cracks are concentrated in the upper part of the specimen where the laser beam was directly irradiated. In addition, it is also found that the laser beam causes thermal damage to both sides of the specimen, and the crack numbers on both sides are similar. Deep cracks can be seen at the front, middle and back of the specimen along the beam scanning direction. From the top view of the rock specimen presented in Figure 11b, the cracks are symmetrically distributed along the central line of the laser moving direction. From the back view of the specimen illustrated in Figure 11c, we can observe that the deepest cracks can reach almost 90% of the specimen's diameter.

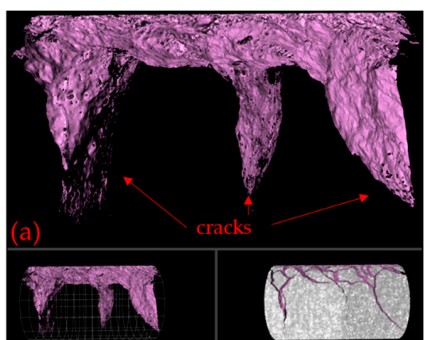 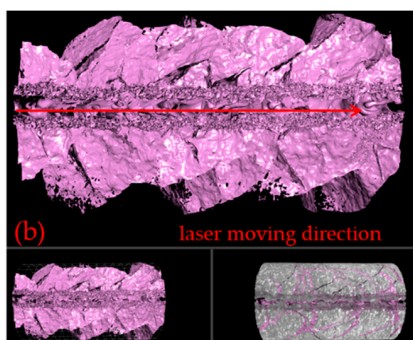 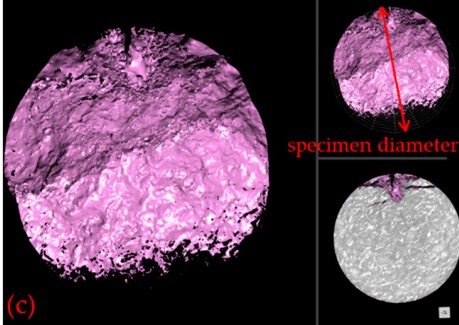

**Figure 11.** Rendered images of irradiated specimens from (**a**) front, (**b**) top and (**c**) right views.

The distribution of cracks within the irradiated specimen can be illustrated further from the vertical-section view of the specimen at different locations, as shown in Figure 12. From the vertical section through the axis presented in Figure 12a, we can clearly see a groove with a uniform depth at the top of the specimen where the laser beam is irradiated directly on the specimen. Deep, cracked gaps are found at the front, middle and back of the specimen as the laser beam moves from the left to the right end. Large and wide cracks can be seen at the same locations of the irradiated specimen, as illustrated in Figure 12b. It is also indicated from Figure 12 that long and wide cracks are present at the two ends of the specimen, which is consistent with the results shown in Figure 11.

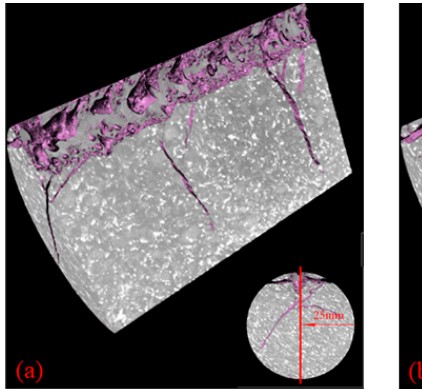
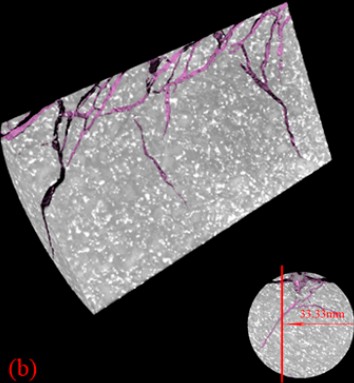

**Figure 12.** Interior-crack images of vertical section at (**a**) axis and (**b**) 3/4 diameter plane.

In order to quantify the crack development within the specimen, the crack length and width at several locations were measured based on the cross- and vertical-section images, as shown in Figure 13. From Figure 13a showing the cross-section located at the beginning of the laser irradiation starting point, the maximum crack width with a value of about 367.48 µm can be found near the bottom of the U-shaped groove generated by the laser beam, where severe damage was created by the high-temperature gradient in the specimen. The width of the cracks decreased as the cracks developed deeper into the bottom. For instance, the width of the cracks on the left and right sides decreased from 330.08 and 367.48 µm to 217.76 and 190.24 µm, respectively. The maximum crack width at the axial section was about 243.98 µm near the right end of the specimen, and it can be observed that cracks at the middle of the specimen are smaller than those near the end, as shown in Figure 13b. It was also found that the lengths of the two cracks at the central and right sides were about 38.27 and 37.93 mm, respectively, from the bottom of the groove to the end of the cracks. In addition, we can also observe that the depth of the U-shaped groove generated by the laser beam is quite uniform and ranges from 11.56 to 12.67 mm, with an open angle (from the bottom, central point to the top of the groove) of about 39°, as shown in Figure 13a,b.

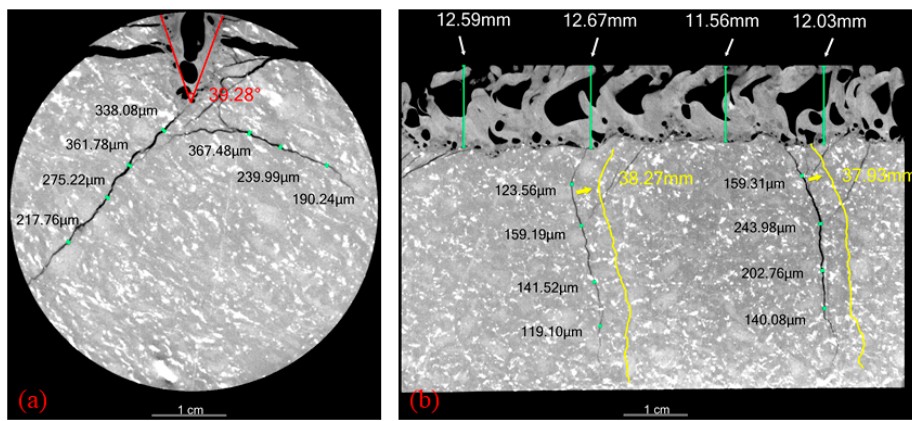

**Figure 13.** Interior-crack length and width at (**a**) starting point and (**b**) axial plane.

## 4. Conclusions

The influence of various laser-power outputs, beam diameter and beam moving speed on the compressive strength, ultrasonic characteristics and crack distributions of the granite specimens were investigated in this paper. The compositions of original and irradiated granite specimens were also quantitatively reported. The results obtained in this study can be used to assess the cracking efficiency of hard granite rock by using laser irradiation, and

therefore provide some useful guidance in the practice. The key findings are summarized as follows:

1. The uniaxial compressive strength of the irradiated graphite is reduced by the higher laser power, smaller beam diameter as well as slower moving speed. The XRD and XRF results indicate that a change from crystalline to amorphous states for the irradiated specimen occurs.

2. Both the ultrasonic-wave velocity and amplitude of the first wave imposed on the irradiated specimens increase with the decreasing laser-power output, increasing the diameter or moving speed of the laser beam. The waveform is also observed to be significantly flattened by the same changes in the irradiation parameters. This suggests that the thermal damage to irradiated specimens caused by laser irradiation can be qualitatively assessed by ultrasonic testing, which is a non-destructive technology.

3. The crack angles and ratio of cracked areas at both ends increase when the laser power increases as well as when the diameter or the moving speed of the laser beam decreases. It is clearly observed that a U-shaped grooving kerf with a depth of about 12 mm is generated, which also matches the beam's movement. A considerable number of cracks are generated around the grooving kerf. Deep cracks are found at the front, middle and back of the irradiated specimen, and the deepest cracks reach almost 90% of the specimen's diameter.

**Author Contributions:** Methodology, Y.W.; investigation, L.K.; data curation, D.Y. and L.S.; writing—original draft preparation, D.Y. and Z.C.; writing—review and editing, J.D. and Y.W.; visualization, D.Y. and L.S.; supervision, Y.W. All authors have read and agreed to the published version of the manuscript.

**Funding:** This study was funded by the National Natural Science Foundation of China (No. 51978653, No. 42107156), Higher Education Discipline Innovation Project (No. B14021), and the Fundamental Research Funds for the Central Universities (No. 2020ZDPYMS18).

**Data Availability Statement:** Data available on request due to the privacy restrictions.

**Conflicts of Interest:** The authors declare no conflict of interest.

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
