# Peer review of "Experimental Investigation on Compressive Strength, Ultrasonic Characteristic and Cracks Distribution of Granite Rock Irradiated by a Moving Laser Beam"

_applsci, doi:10.3390/app122010681_

Round 1
Reviewer 1 Report
The manuscript “is aimed to investigate the influence of the laser power output, beam diameter and moving speed on the mechanical property, ultrasonic characteristic and crack distribution of the irradiated granite samples.” The results are interesting, however the quality of writing needs considerable improvement.
If by “mechanical property” the authors mean UCS, it should be said so from the very beginning.
Please explain what “original” means - samples without laser treatment?
Comments on the effect of laser treatment on the failure patterns in uniaxial compression are necessary.
Abstract: What “The ultrasonic wave velocity and amplitude of the first wave are both increased by a decreased laser power, …“ means? Something is increasing with decreasing laser power - should it be decreasing with increased laser power? After all, you do not have to apply laser.
Line 55: “save the construction cost.” - you either reduce construction cost or save on the construction cost.
Line 92: “The study on the mechanical property, ultrasonic characteristic and crack distribution in the laser-irradiated granite using a moving beam has not yet been reported.” - should be “A study…”.
Line 103: Diameters are not denote by Greek letter phi.
Section 3.1 - should be Mechanical Properties.
Line 173: “Therefore, it suggests that a lower moving speed should be taken into account to achieve a better thermal damage on rock samples.” - Should be “Therefore, it suggests that a lower moving speed should be used to achieve a better thermal damage on rock samples.”
Section 3.2. Ultrasonic Characteristic - should be 3.2. Ultrasonic Characteristics
Section 3.3.1. Surface Crack - should be 3.3.1. Surface Cracks
Section 3.3.2. Interior Crack - should be 3.3.2. Internal Cracks. The same should be corrected in the text.
English should be checked by the authors. The main problem is that the authors do not recognise plural, using singular instead. The companies offering checking English will not be able to correct this as they would not necessarily understand the technical meaning.
Author Response
Reviewer #1:
The manuscript “is aimed to investigate the influence of the laser power output, beam diameter and moving speed on the mechanical property, ultrasonic characteristic and crack distribution of the irradiated granite samples.” The results are interesting, however the quality of writing needs considerable improvement.
(1) If by “mechanical property” the authors mean UCS, it should be said so from the very beginning.
Response: Thanks for your kind suggestion. In the revised manuscript, "uniaxial compressive strength " was used to replace "mechanical property".
(2) Please explain what “original” means - samples without laser treatment?
Response: " original samples" means the specimens without laser irradiation, which was explained in the Abstract and the text.
(3) Comments on the effect of laser treatment on the failure patterns in uniaxial compression are necessary.
Response: The failure patterns of granite specimens were added in the revised manuscript. The details are:
When the uniaxial compression stress approached to the peak, the granite fractured instantaneously accompanied by sounds similar to the rock burst. A considerable number of cracks penetrating from the top to bottom were observed when the uniaxial compression load approached to the yield strength of the original rock specimen, as shown in Figure 4. A typical splitting failure with a slight deformation can be confirmed for the original granite exposed uniaxial compression. This can contribute to that the granite is a kind of rock with high brittleness. The same cracks can be seen from the irradiated specimens shown in Figure 4. However, the deformations are considerably increased when the laser power increases, and a detailed discussion is presented in section 3.1.2. In addition, the desquamation and fragments' falling from the irradiated specimens are observed during the uniaxial compressive strength test, which is due to the thermal damages caused by the laser irradiation. Figure 4 also indicates that the failure modes of irradiated specimens are similar under various irradiation conditions. The destruction of the rock specimen is attributed to the internal micro-cracks' initiation, propagation and penetration under the uniaxial compressive load.
(4) Abstract: What “The ultrasonic wave velocity and amplitude of the first wave are both increased by a decreased laser power, …“ means? Something is increasing with decreasing laser power - should it be decreasing with increased laser power? After all, you do not have to apply laser.
Response: Thanks for the comment. This sentence was change to: "The ultrasonic wave velocity and amplitude of the first wave are both increasing with a decreased laser power, increased diameter or moving speed of laser beam."
(5) Line 55: “save the construction cost.” - you either reduce construction cost or save on the construction cost.
Response: "save the construction cost" was replaced by "reduce the construction cost" in the revised manuscript.
(6) Line 92: “The study on the mechanical property, ultrasonic characteristic and crack distribution in the laser-irradiated granite using a moving beam has not yet been reported.” - should be “A study…”.
Response: "The study on …" was change into "A study on …" based on the suggestion.
(7) Line 103: Diameters are not denote by Greek letter phi.
Response: The diameter of the rock specimen was denoted by Greek letter φ.
(8) Section 3.1 - should be Mechanical Properties.
Response: The title of Section 3.1 was changed into "Mechanical Properties".
(9) Line 173: “Therefore, it suggests that a lower moving speed should be taken into account to achieve a better thermal damage on rock samples.” - Should be “Therefore, it suggests that a lower moving speed should be used to achieve a better thermal damage on rock samples.”
Response: This sentence was also revised based on the reviewer's suggestion in the revised manuscript.
(10) Section 3.2. Ultrasonic Characteristic - should be 3.2. Ultrasonic Characteristics
Response: The section title of "Ultrasonic Characteristic" was substituted by " Ultrasonic Characteristics ".
(11) Section 3.3.1. Surface Crack - should be 3.3.1. Surface Cracks; Section 3.3.2. Interior Crack - should be 3.3.2. Internal Cracks. The same should be corrected in the text.
Response: In the revised manuscript, "cracks" was used to replaced "crack" in Section 3.3. The same was conducted in the Abstract and Conclusion.
English should be checked by the authors. The main problem is that the authors do not recognise plural, using singular instead. The companies offering checking English will not be able to correct this as they would not necessarily understand the technical meaning.
Response: Thanks for your kind suggestion, the revised manuscript was totally checked. The plural form including cracks, characteristics etc. was used to replace their corresponding singular one.
Once again, we greatly appreciate editor and reviewers for your time to review the submitted manuscript and providing such positive feedback and useful suggestions, and hope that the responses are sufficient and appropriate.

Reviewer 2 Report
Dear Authors,
You have done a lot of efforts the capturing the results. Nevertheless, some corrections are needed. Please follow the article with comments and suggestions.
Reviewer

Author Response
Reviewer #2:
- The use of the name mechanical properties in the title is not covered by the text of the article.
Response: Thanks for your kind suggestion. In the revised manuscript, "compressive strength" was used to replace "mechanical property".
- Please add more details i.e. diameter of ... etc.
Response: This sentence was changed into: "…dependent on the power, diameter and moving speed of laser beam".
- This sentence follows summary.
The results achieved in this study can be used to assess the cracking efficiency of hard granite rock by using the laser irradiation and therefore provide some useful guidance in the practice.
Response: Based on your suggestion, this sentence was removed from the Section of Introduction and put into the Section of Conclusion.
- What exactly means the standard: ASTM, EN, BSSM etc. ?
Response: The China Standard GB/T50266-2013 was added in the revised manuscript.
- A scheme or details in a form of balloons is needed for better analysis.
Response: The experimental program was added in the revised manuscript.
Table 1. Experimental scheme.
Parameters |
I |
II |
III |
IV |
Laser power (W) |
400 |
600 |
800 |
1000 |
Laser beam diameter (mm) |
6 |
8 |
10 |
12 |
Moving speed of laser beam (mm/s) |
0.5 |
1.0 |
2.0 |
4.0 |
- Accuracy follows force or/and displacement, therefore please select one or both physical magnitudes.
Response: Thanks for the suggestion. In the revised manuscript, one physical magnitude for the force of the electro-hydraulic testing servo machine was selected.
- Please add the full name of the equipment
Response: The full name of the equipment was added in the revised manuscript.
- Which kind of uncertainty did you select?
Response: The details of the uncertainty analysis was added:
The testing accuracy and uncertainty were analyzed to ensure the accuracy of the experimental results. The equations of uncertainties for directed variables including rock mass, length etc. are present by [Error! Reference source not found.]:
(1)
where uv is uncertainty of directed variables, â–³v is the testing accuracy, σv is the Bessel equation of standard deviation which is shown:
(2)
where xi and are individual testing values and the mean value of individual testing values respectively, N is the number of testing items.
The equations of uncertainties for undirected variables (including the volume, uniaxial compressive strength) are shown as [Error! Reference source not found.]:
(3)
(4)
where is undirected variable calculated from xi. The Equation (3) could be used to compute the uncertainties if the F(xi) just includes operators of add and subtract. Equation (4) should be employed if the F(xi) just includes operators of multiplication and division.
- Do you have the same details on values of temperature after the laser process?
Response: The rock temperature was measured by using an infrared camera with the maximum temperature testing capacity up to 2000 oC. The effect of laser irradiation parameters on the rock temperature was also experimentally investigated in another published paper. However, the analysis on the rock temperature was not presented in this study to avoid the repetition with our previous paper. If the reviewer is concerned about the rock temperature, some details could be supplied by the published paper in the journal of Renewable Energy (DOI: 10.1016/j.renene.2020.06.138).
- Why would you like to use 2 decimal places for the values of the ultimate compressive strength?
Response: In the revised manuscript, only the integer part was presented for the compressive strength.
- Some comments on specimens' temperature during the compression test should be added.
Response: Thanks for your kind comments. The aim of this manuscript is to investigate the effect of laser irradiation on the compressive strength, ultrasonic characteristic and cracks distribution. The rock temperature induced by laser irradiation was tested by using an infrared camera (Flir -SC655, USA). The experimental investigation on the influence of laser parameters on the rock temperature was also conducted. However, the temperature variation during the compression test was not studied in this manuscript. The authors acknowledge that the rock temperature during the compression test is very interesting, and appreciate the reviewer's suggestion. We will conduct this investigation in the near future.
- These photos should be presented in another Figure
Response: Thanks for your kind suggestion. The photos in Figure 3 were removed and presented in Figure 4 in the revised manuscript.
- Please add compressive curves with the value of mechanical parameters for the rock before and after laser acting.
Response: The stress-strain curve for the original specimen and irradiated granite with different irradiation parameters was presented in Figure 3d. One can find that the compression distance of the irradiated specimens was longer than that of the original rock. The strains corresponding to the peak stress of the irradiated specimens were dependent on the irradiation parameters, which were bigger than that of the original one.
Figure 3. (d) uniaxial compressive strength-strain curve.
- This text can be subjected to analysis if the photos showing the material degradation will be presented at better quality.
Response: These photos were presented in another figure based on the suggestion.
- Would you like to add more details on the temperature measurement?
Response: The rock temperature induced by laser irradiation was tested by using an infrared camera (Flir-SC655, USA). Due to the detailed investigation on the rock temperature induced by laser irradiation, the temperature measurement was not provided in this study. If the reviewers and the readers are concerned about the temperature measurement, some previous published papers could be referenced.
(DOI: 10.1007/s00231-019-02682-2, 10.1016/j.renene.2020.06.138).
- This text should replaced to introduction
Response: This text was removed from the Section 3.1.1 based on the suggestion.
- With respect to very small photos showing specimens after mechanical tests, this text is possible to analyse after the figures reach a better quality.
Response: Thanks for your suggestion. These photos were presented in additional figures in the revised manuscript (Figure 8 and Figure 9).
- The same comments as above are addressed to the photos i.e. additional figure is required.
Response: These photos were presented in additional figures in the revised manuscript (Figure 8 and Figure 9).
- Please connect the details presented in this figure with the description, adding balloons or additional words.
Response: Additional words were added in the Figure 11 to connect the details presented in these figures with the description.
Figure 11. Rendered images of irradiated specimen from (a) front, (b) top and (c) right view.
- A scheme for a specimen after the laser acting is needed.
Response: The process for the 3D-XRM test was added in the revised manuscript based on your suggestion. the details are:
CT scanning technology can visually present the microstructure characteristics (such as cracks, pores, micro-cracks, etc.) of the irradiated rock by using 256 gray scales through the density difference of each imaging unit in the specimen. Afterwards, Dragonfly is applied for post-processing, and data such as crack sizes inside the irradiated sample are obtained by means of smoothing filtering, threshold segmentation, mesh reconstruction and size measurement.
Once again, we greatly appreciate editor and reviewers for your time to review the submitted manuscript and providing such positive feedback and useful suggestions, and hope that the responses are sufficient and appropriate.

Round 2
Reviewer 2 Report
Dear Authors,
Your paper follows a high scientific level. I recommend it for publication in Materials.
Reviewer